# Stress and Health Outcomes in Midwestern Latinx Youth: The Moderating Role of Ethnic Pride

**DOI:** 10.3390/ijerph192416966

**Published:** 2022-12-17

**Authors:** Blake L. Jones, Matthew K. Grendell, Joshua M. Bezzant, Keeley A. Russell, Brooke W. Williams, Lainey Jensen, Carli Peterson, Joshua Christensen, Brynn Pyper, Jaren Muh, Zoe E. Taylor

**Affiliations:** 1Department of Psychology, Brigham Young University, 1092 KMBL, Provo, UT 84602, USA; 2Department of Human Development and Family Science, Purdue University, West Lafayette, IN 47907, USA

**Keywords:** depressive symptoms, externalizing behaviors, self-esteem, culture, Latinx, ethnic pride, children, stressful events, family, obesity

## Abstract

Background: Stress has been linked to numerous health outcomes, including internalizing and externalizing behaviors, self-esteem, and physical health. Culture has also been linked to stress and health. This study examined the links between stress and health, and the potential moderating role of Latinx ethnic pride (LEP). Methods: The sample consisted of 119 Latinx youth from the Midwestern U.S. Mothers and youth completed surveys. Variables included the Multicultural Events Scale for Adolescents (MESA), parent and home stressors/risks (PHSR), LEP, depressive symptoms, aggression, frustration, and self-esteem. Research assistants measured child heights and weights and calculated BMI percentiles. Results: LEP was negatively related to MESA, depressive symptoms, aggression, and frustration, and positively related to self-esteem. MESA and PHSR were associated with depressive symptoms, aggression, frustration, and self-esteem, but not with BMI percentile. In adjusted regression analyses, LEP moderated the effects MESA had on frustration and self-esteem, marginally moderated the link between MESA and depressive symptoms, and was not related to aggression or BMI percentile. LEP did not moderate the relationship between PHSR with any health outcomes. Conclusions: Stressors were generally related to child mental health. LEP may play an important role in protecting against some of the effects of stressful events on mental health outcomes.

## 1. Introduction

The human body is in a constant process of maintaining a dynamic equilibrium known as homeostasis [1]. Sometimes, life events, environmental conditions, or other phenomena can knock this equilibrium out of place and activate the body′s natural stress responses. These phenomena are known as stressors. With this, an individual experiences “stress” anytime a stressor is present that challenges or threatens to challenge homeostasis [1]. In response to stress, the body employs a chain of hormones through both the sympathetic nervous system (SNS) and the hypothalamic-pituitary adrenocortical (HPA) axis [2]. Through this process, the body creates epinephrine through the SNS and cortisol through the HPA axis, which then work to create and reroute energy to the critical systems of the body [2]. While these processes help the body deal with stressors in the short term, when these systems are active for too long or too often due to chronic stressors, significant damage can be done to the body [2].

In addition to the SNS and the HPA axis, the body also uses adaptation to better respond to stressors and maintain homeostasis through a process called allostasis [3]. Often, this may be achieved through an unhealthy use of a high level of stress hormones, and the cumulation of the consequent changes is called allostatic load. Allostatic load significantly impacts the body and, in the long term, can cause harm to the body [1,3]. In particular, it would appear that adolescents are especially vulnerable to these detrimental effects of stress, which may be because adolescents′ brains are still maturing, they are experiencing greater hormonal reactivity, and they may experience increased corticosterone sensitivity [4]. It would appear that adolescence is a critical developmental period for stress [1]. Strong and chronic stressors during this time period may lead to lifelong allostasis [1], long-term impacts to one′s mental or physical health, and even death [2].

### 1.1. Stress and Mental Health

Along with the impacts of chronic stress, general stress may also impact mental health, especially with regard to depression, self-esteem, and externalizing behaviors [5]. According to the National Institute of Mental Health, approximately 17% of adolescents in the United States ages 12 to 17 experienced at least one major depressive episode during 2020 [6]. In an attempt to identify factors of depression, studies [5,7] have found that this prevalent form of psychopathology appears to be heavily correlated with one′s stress levels. Specifically, various stressors can also predict depressive symptoms in youth, such as stressful life events [8] and parent–child conflicts [9]. Currently, there are multiple theories that suggest why this relationship between stress and depression may exist. Stress levels are positively correlated with proinflammatory cytokine levels, which in turn are connected to depressive symptoms [2]. These researchers suggested that elevated cytokine levels could mediate the relationship between stress and depression. Other researchers found another mediatory relationship in which stressful life events predicted more use of disengaged coping, which predicted, in turn, depressive symptoms [8]. They proposed that not only may stress itself be a factor when it comes to depression, but also the way in which an individual chooses to react to and cope with stressors.

Externalizing behaviors, including frustration and aggression, have also been associated with increased stress [10,11]. For example, family stressors such as conflict and instability have been associated with externalizing behaviors in adolescence [12,13]. Because adolescence is a time of autonomy-seeking and identity-formation, sources of stress can come from family, friends, and school. During adolescence, some youth may have a difficult time processing the various stressors that they experience, leading to an increased risk for reacting in unhealthy ways such as exhibiting externalizing behaviors [14,15]. 

In addition to depression and externalizing behaviors, stress also has been connected to lower self-esteem [16,17]. Specific stressors, such as family violence, family alcohol abuse, [18] and parent–child conflict [9], also show connections to lower self-esteem in some youth. These findings would suggest that stress is connected to one′s self-esteem, though other variables may influence or weaken that relationship, such as gender [18].

### 1.2. Stress and Physical Health

Stress has also been shown to have detrimental effects on physical health, both in the long term and in more acute situations. Because the stress response invests energy into prepping the parts of the body necessary to cope with a stressor, the body activates other high-stress functions to take action against threats, such as increasing blood flow and strengthening the immune system [2]. While these outcomes are important for assisting the body to cope with acute stress, they can prove unhealthy over time. During times of chronic stress, having the physiological stress-response systems activated for long periods of time has been shown to be harmful to physical health, leading to such problems as chronically elevated blood pressure and weakened or damaged immunity [2]. Increased blood pressure from stress is linked to cardiovascular difficulties in particular [2], and stress has also been linked to the development of autoimmune disorders [19]. Over time, the stress response can prove unhealthy and dangerous to the body, especially with longer periods of stress.

One well-documented outcome of stress is obesity, which has multiple negative health outcomes as well. Stress has been linked to obesity through stress-induced disrupted brain function, hormone production, a lack of exercise [20], and sleep deprivation [21]. Poor self-regulation and cumulative risks in childhood are correlated with obesity later in life [8], adding to research that links adverse childhood experiences with exacerbated stress responses [22], showing that childhood could be a critical period for developing habits that could alleviate later risks for stress and obesity. Obesity can lead to other health problems, including hypertension, diabetes, breathing problems, and heart disease [23]. While obesity is a widespread and significant threat to physical health and a major financial burden [24], the connection obesity has to stress could be particularly notable in certain populations.

### 1.3. Culture and Stress

Acculturation and acculturative stress are involved in many aspects of life in a foreign culture. Language and intergenerational conflict are primary stressors in acculturation, particularly within first- and second- generation families [25], but ethnic identity and cultural orientation have shown to be mediators and sources of resilience in acculturative stress [25,26]. Additionally, differences in acculturation in a household has been linked to other intergenerational conflict [27], and differences in endorsed culture also predict lower self-esteem and higher aggression in children [28]. Acculturation appears to influence stress in several different domains of life due to the complicated nature of culture and identity when living and adjusting to a foreign culture. Acculturative stress in adolescents has also been related to anti-social externalizing behaviors [11].

It is known that acculturative stress, as well as allostatic load, is a trend not only found amongst Latinx individuals, but also within Latinx individuals across time. Comparing U.S.-born Hispanics to non-U.S.-born Hispanics shows that those born in the U.S. have a higher allostatic load, while those who immigrated show a positive correlation between allostatic load and time in the U.S. [29]. A similar trend is found in obesity risk levels in Latinx immigrants. Numerous studies have shown that Latinx foreign-born adults have a lower risk of obesity compared to U.S.-born adults; however, the longer they reside in the U.S., the more the risk of obesity increases [30,31,32,33]. Although it is known that there is an uptrend in BMI among Latinx adults [23] and that prevalence among Latinx youth shows 26.2% of Hispanic children are obese according to the Centers for Disease Control Prevention [34], additional studies show a more concrete link between stress and obesity. This is especially concerning because the number of Latinx adults who are classified as obese is now approximately 45% [35]. Allostatic load was shown to be associated with a higher risk of obesity among Latinx adults and children [29]. Stress is shown to increase the risk of obesity, especially in Latinx individuals.

### 1.4. Culture and Mental Health

There could be many factors that influence individuals living in a foreign culture that affect their health for various reasons. Previous research has established links between culture and both physical and mental health. For example, Latinx adolescents show higher acculturative stress than non-Latinx adolescents [28]. As previously mentioned, stress can have detrimental effects on a person [2], but the type of stress that accompanies living in a foreign culture, or acculturative stress, has been studied in depth for health-related outcomes [36].

Concerning how culture is related to mental health, there are multiple aspects to be considered, one of which is acculturation. Acculturation is generally defined as assimilating to a different culture, which is usually the dominant culture in that location. Numerous studies have discovered correlations between acculturation and depression [7,25,36] or self-esteem [26]. This relationship between acculturation and mental health continues to appear when the acculturation occurs within the context of the family. For example, the connection between acculturation and depressive symptoms was stronger when both the parent and child reported experiencing acculturative stress [11]. Acculturative stress would be described as the negative stressors or health outcomes that are experienced by individuals who are going through acculturation. This association between acculturation and mental health appears to be particularly present with acculturative conflict in the family. When there is conflict between generations on the level of acculturation, individuals experience worse self-esteem [26] and greater depression [25,37]. In addition, acculturation at times can reflect differences in adherence to cultural values. Some researchers found no direct relationship between acculturation and self-esteem, although they did report that differing cultural values between parents and children were associated with lower self-esteem [38]. These findings indicate the important role that family can play in the relationship between culture and mental health.

In addition to acculturation and familial aspects of culture, the social factors of culture are also connected to one′s mental health. When looking at the impacts of discrimination, several studies have connected this social aspect of culture to low self-esteem and depression [9,39,40,41]. The connection to depression continues to hold, even when controlling for other potential stressors [42]. Similar findings connected cultural invalidation with depression [43]. Together, these findings highlight a possible connection between culture′s social manifestations and an individual′s mental health.

Ethnic identity may also play a role when it comes to one′s mental health; however, there appears to be some disagreement in the current literature. While some researchers have found that a strong ethnic identity is associated with less depression [25] and that poor ethnic identity is correlated with increased depression [44], others have found no connection between ethnic identity and depression [42,45]. Ethnic identity does, however, appear to influence adjacent factors. For example, despite not finding a connection between ethnic identity and depression, researchers found that stronger ethnic identity was correlated with improved personal recovery from depressive symptoms [45]. This would suggest that while findings concerning the relationship between ethnic identity and mental health are contradictory, having a strong ethnic identity may at least better allow individuals to recover from their symptoms of psychopathology.

### 1.5. Culture and Physical Health

Possible relationships between culture and health extend beyond mental health and into physical health. Of particular interest to our study is the way culture may impact obesity and overweight. The association between culture and the physical health outcomes of being overweight and obese has been confirmed by a number of studies, though the conclusions regarding the relationship are not unanimous. In the majority of these studies, measures or proxies of acculturation are used to demonstrate the ways in which changes in the cultural orientation of a Latinx individual are related to adiposity or BMI. By using acculturation measures, these researchers can identify which of the two cultures involved in acculturation is more correlated with obesity and overweight risks. In the population of interest to our study, the cultures under examination are the mainstream culture of the United States (U.S.) and the cultures of origin for Latinx immigrants and their descendants.

A consideration of physical health as a whole and its relationship with culture has been pursued by fewer studies. The two studies we were able to find depended on participants reporting their perceptions about their own physical health. One of these studies discovered that greater acculturation among Latinx women was associated with perceptions of better physical health [46], while the other found that more acculturated Latinx elders reported no better physical health than less acculturated Latinx elders [47]. Despite their differing populations of interest, these studies offer contradictory findings with regard to Latinx individuals′ personal perceptions of physical health and their level of acculturation.

A few studies have found that greater acculturation does not correlate with higher risks of obesity and overweight for Latinx adolescents, though these studies employed relatively simple acculturation proxies. One found that Mexican-born adolescents in the U.S. did not have lower overweight or obesity risk compared to U.S.-born Latinx adolescents, but nativity and time spent in the U.S. were the only acculturation proxies used to reach this conclusion and offer very little insight into the actual cultural orientation of the individual [48]. The other study found that Latinx adolescents who were recent immigrants were actually more likely to be obese than their U.S.-born counterparts, but no other measurement than status as an immigrant or a native was used to reach this conclusion [28]. These studies are more recent than all but two of the studies that found greater acculturation to be associated with a greater risk or presence of obesity and overweight [49,50], and so may reflect a recent change in trends. Despite this possibility, these two recent studies may provide greater insight into the relationship between culture and overweight or obesity because they evaluated acculturation levels using several more culturally relevant factors than time spent in the U.S. and nativity.

Most studies that consider the relationship between the acculturation of Latinx adolescents and the adiposity or BMI of the same population conclude that an increase in orientation toward the mainstream U.S. culture is associated with an increase in the risk or presence of obesity and overweight [30,31,33,49,51]. Obesity risk was associated with acculturation through a number of factors, including the location to which one immigrates and one′s age at immigration [50]. This possible relationship can be understood by considering that younger immigrants are more likely to adopt the mainstream U.S. culture than older immigrants [9,44,47] and that some neighborhoods will be more oriented toward the mainstream U.S. culture than others [32,52]. The obesity side of the relationship is generally attributed to the mainstream U.S. diet being more obesogenic than the typical diet of Latin American countries [32,48,51], meaning that as Latinx youth orient themselves more toward mainstream U.S. culture, they are also orienting themselves towards a diet that is more likely to cause overweight and obesity.

It has been found that integrated youth, or youth that still have contact with their heritage, have healthier eating habits. Those who have assimilated and no longer accept their heritage culture tend to have worse eating habits [52]. These bad eating habits are a result of many things, including a low socioeconomic status, marketing messages, and parenting. Oftentimes, advertisements made for the Latinx population are filled with calorically dense foods and drinks [45]. With a lack of accessibility to healthy foods due to their socioeconomic status and advertisements that celebrate foods with little nutritional value, the Latinx community suffers great health detriments.

As was previously mentioned, diet has been identified as a possible explanation for the relationship between acculturation and physical health outcomes. This suggestion is due, at least in part, to the way acculturation impacts diet itself. Some studies have found that less acculturated Latinx individuals have healthier diets than those who are more acculturated [53,54]. Other studies make the relationship more apparent by pointing to higher intake of fruits and vegetables in communities that are oriented more towards Latinx cultures of origin rather than the U.S. mainstream culture [51] and lower intake of sodium and empty calories by Latinx youth who were considered integrated in terms of acculturation [52]. These healthier diets are more likely to protect against being overweight and obese. The relationship between culture and obesity or overweight in Latinx individuals is not as clear as it might seem. In addition to the studies that challenge the conclusion that greater orientation to mainstream U.S. culture is related to greater obesity and overweight risks, other studies have found possible contradictions within the theory that the stated relationship is due to differences in diet.

It was also found that Latinx youth who were considered as separated (completely oriented toward their culture of origin) or marginalized (not oriented toward either mainstream U.S. culture or their culture of origin) had diets that were higher in empty calories and lower in whole grains than integrated and even assimilated (completely oriented toward mainstream U.S. culture) Latinx youth [52]. This finding suggests that an orientation toward the culture of origin is not entirely responsible for the better dietary practices of Latinx youth.

One additional study found that there was no dietary advantage for U.S. immigrants from Mexico, pointing to the Mexican diet becoming more obesogenic [48]. An older study also points to changes in nutrition being associated with increasing obesity prevalence in Latin American countries [55]. Some researchers suggest that, due to the Mexican diet becoming more like the U.S. diet in recent years, in terms of increasing the risk of obesity, Mexican immigrants would have the same obesity risk whether they oriented themselves toward the diet promoted by mainstream U.S. culture or toward the diet promoted by their culture of origin [48].

### 1.6. The Role of Culture as a Moderator

Not only can culture and stress individually play roles in affecting one′s health, but Latinx culture in general can act as a moderator in the relationship between stress and health. This relationship appears to be particularly salient among youth, as they are in a period of identity exploration and formation [56]. However, there does appear to be some variation in how Latinx culture moderates the relationship. For example, when Latinx individuals were more acculturated to the United States′ culture, they self-reported better mental health outcomes [47]. This directly contradicts other findings that a stronger ethnic identity protects against poor mental health from acculturative stress [25]. This discrepancy may be because of one′s stage of identity exploration, seeing as discrimination has more negative impacts on an individual who is exploring their ethnic identity whereas a commitment to one′s ethnic identity can instead be protective against discrimination′s negative effects [57]. All these pieces of evidence highlight that culture can play both a positive and negative role in moderating the relationship between stress and mental health.

As previously established, stress is connected to depression [2,5,7,8,9], but different aspects of culture may weaken this relationship. For example, both ethnic pride [58] and positive ethnic identity [40] are correlated with lower depression. Further, bicultural ethnic identity appears to show a similar correlation with less depressive symptoms [9] while also mediating the relationship between discrimination and depression [41]. In addition to how differing forms of ethnic identity can influence the relationship, cultural values such as familism may also moderate the relationship, even to the point of nullifying the negative impacts of acculturative conflict [7,37]. Altogether, these findings illustrate some of the many ways in which culture can positively impact the connection between stress and depression.

Aside from how culture may moderate the relationship between stress and mental health, culture may also influence the behaviors that impact this same relationship. Ethnic pride can also protect against poor health behaviors such as smoking or risky sexual behavior among adolescents [59]. High ethnic pride also may reduce the amount of alcohol consumed by those receiving cognitive behavioral therapy for substance abuse, thus improving the effectiveness of treatment [60]. Strong ethnic identity has also been linked to the ability to recover from serious mental illness [45]. These impacts would indicate that recovery and behaviors can be influenced by cultural factors, which in turn can help alleviate mental illnesses or protect against future potential health complications.

While these findings highlight the many positive moderating impacts of culture on the relationship between stress and health, there are also many negatively moderating impacts to consider. Researchers found that bicultural ethnic identity was associated with higher self-esteem [9], but others reported that when a low bicultural ethnic identity was present, acculturation conflict predicted lower self-esteem [26]. Similarly, differences in cultural values between parents and children may also be associated with lower self-esteem [38].

In addition to impacting self-esteem, strong ethnic identity can exacerbate the negative impacts of stressors on health. As previously established, discrimination may be linked to depression [9,39,40,41], but an interesting interaction appears when ethnic identity is considered within the context of this relationship. A strong ethnic identity can worsen discrimination′s relationship with depression [39,40] and minority stress [42]. This may be because a stronger ethnic identity influences individuals to identify more with their culture and become more sensitive to discrimination [40,42]. A similar theory posits that those with lower levels of bicultural identity are associated with increased discrimination [41], which could in turn lead to correlated mental health issues.

### 1.7. Acculturation Varies

Typically, an individual′s level of acculturation is considered as belonging within a spectrum as opposed to within a binary system of individuals being either acculturated or not [26,27,52]. The spectrum exists because of the assumption that individuals progressively adopt some aspects of the host culture while also retaining or jettisoning aspects of their culture of origin. This process of acculturation is distinct for every individual in terms of initial levels of acculturation, the pace at which it occurs, and the extent to which it occurs. It is this variety within the process that gives rise to the spectrum of acculturation.

Various studies have found that the extent to which a Latinx immigrant acculturates to the U.S. culture is influenced by factors such as an individual′s age at the time of immigration and the community into which they enter [26,32,38,50,51,52]. Additional studies have discussed how the rate at which a Latinx individual acculturates in the U.S. is impacted by similar factors as those just mentioned [27,46,47]. Therefore, as circumstances vary, so does acculturation. Studies have also found that a Latinx individual′s original nationality impacts acculturation, altering the ways in which acculturation serves as a moderator between measured variables [32,61,62]. These findings can be understood to point out that even within levels of acculturation there are differences with regard to how acculturation impacts the individual. These considerations suggest that differences in acculturation can exist within not only communities but also families, with differences such as nationality, age, neighborhood, peers, etc., all impacting the level and rate of acculturation for each individual. These differences in acculturation are meaningful because a Latinx individual′s level of acculturation has been found to impact his or her mental and physical health outcomes [9,29,32,47,49,53,63,64,65].

Guided by the previous literature, the current study examined the links between stress and risks that may be prevalent in the homes and lives of Latinx youth in the Midwest and several health outcomes, including internalizing (depression) and externalizing (aggression and frustration) behaviors and self-esteem, as well as physical health (obesity). In addition to assessing these main effects, this study also assessed the potential role of culture by testing whether ethnic pride would moderate the connections between stressors and health outcomes. Based on previous research, the authors made the following hypotheses: H1—stressful life events would be related to all five of the health outcomes; H2—parent and home stressors would be directly related to all five of the health outcomes; H3—Ethnic pride would moderate the links between stressful life events and health outcomes; and H4—Ethnic pride would moderate the links between parent and home stressors and all five of the health outcomes in Latinx youth in the Midwest.

## 2. Materials and Methods

### 2.1. Participants

Participants included 119 Latinx preadolescents (59% girls), approximately aged 10–12 years (*M* age = 11.53, *SD* = 0.69), living in the Midwestern United States. Most youth were born in the U.S. (95%), although most of the parents reporting being born outside of the U.S. (95%, almost all in Mexico). Parents had lived in the U.S. for an average of 16.44 years (*SD* = 5.59 years). Most families reported having two parents in the home (88%), and the majority of parents (69% of mothers, 79% of fathers) had not completed high school. Annual family income was between $25,001 and $30,000, and most families (78%) lived in rural Midwestern towns with populations of fewer than 20,000 people. The other 22% of families lived in a midsized city of around 70,000 people.

### 2.2. Procedure

The study was approved by the Purdue University IRB. Inclusion criteria included a willingness of both the child and mother to participate and their self-identification as Latinx. Participants were recruited from local and surrounding communities through fliers distributed by schools and extension educators, referral sampling of other participants, and local advertisements. Interested families contacted the research team via phone or email and then completed a brief screening questionnaire. After determining eligibility, participants were visited in their homes by trained bilingual research assistants (RAs). The youth were measured for height and weight by trained research assistants and were given surveys to complete on their own in their homes. Additionally, they were given instructions for completing the surveys and large sealable envelopes to put the surveys in when they were completed. Researchers returned to the homes approximately three days later to pick up the surveys, answer any questions about the surveys or study, and pay participants. Mothers were paid $50 for participation, and children were paid $40 for participation.

### 2.3. Measures

**Demographic Data**. Mothers completed longer surveys that included questions about the family and child, including child birthdate, parent education, family income levels, and other questions about the home and family environment.

**Mental Health Outcomes**. Youth completed the EATQ-R, which is a 19-item survey developed by Capaldi and Rothbart [66]. These items form subscales for internalizing (***depressive symptoms***; 6 items with Cronbach′s α = 0.75) and externalizing (***aggression*,** 6 items with Cronbach′s α = 0.77, and ***frustration***, 7 items with Cronbach′s α = 0.75) behaviors. Participants answered the EATQ-R using a Likert-type scale ranging from 1 (almost always false) to 5 (almost always true). ***Self-esteem*** was completed by youth about themselves as well, using the 10-item Rosenberg Self-Esteem Scale [67,68], with a Cronbach′s α = 0.78.

**Physical Health—Child BMI Percentile**. Two trained RAs measured and weighed each participant (children and parents) in their homes using a portable SECA digital scale and SECA stadiometer using standard protocols from the Center for Disease Control. Child BMI percentiles were calculated using the CDC′s “Children′s BMI Tool for Schools” calculator [69]. This webtool adjusts child BMI percentiles by child sex and age.

**Latinx Ethnic Pride** (**LEP**). Adolescents completed an 8-item scale regarding their level of pride and connection to belonging in the Latinx culture. These questions were adapted from Thayer et al. [70] and Phinney [71] and have been used in various studies of Latinx American adolescents [72,73]. Items included the following statements: 1. You have a lot of pride in your Latinx roots. 2. You feel good about your cultural and ethnic background. 3. You like people to know that your family is Latinx. 4. You feel proud to see Latinx actors, musicians, and artists being successful. 5. You are active in organizations or social groups that include mostly Latinos. 6. You are happy that you are Latinx. 7. You participate in Latinx cultural traditions such as special food, music, or customs. 8. You feel a strong attachment towards your own ethnic group. Participants responded using a Likert-type scale ranging from 1 (strongly disagree) to 5 (strongly agree), and the scale reliability was high, with a Cronbach’s α = 0.91.

**Multicultural Events Scale for Adolescents** (**MESA**). This was a 35-item scale that asked adolescents to answer whether certain stressful events had happened in the past three months. They answered yes or no to each item, such as, “During the past 3 months, your parent lost a job”; “During the past 3 months your family had to stay in a homeless shelter or public place”; “You were pressured to do drugs, smoke, or drink alcohol”; and “A close friend had a serious emotional problem.” A sum score was then taken to represent having experienced more life-event stressors. This scale was taken from Gonzales et al. [74].

**Parent and Home Stressors and Risks** (**PHSR**). This variable was composed of four parent or family stressors or risks within the home that have been linked to adolescents′ behaviors. These variables were all taken from questionnaires. The first variable was child perception of interparental conflict (CPIC), which included 14 items [75]. In addition, three other variables were taken from mothers′ questionnaires about themselves, including mother-reported work–family conflicts (10 items, with 5 work–family conflicts and 5 family–work conflicts), maternal depression [66], and maternal anxiety [76]. Each of the items on this scale was cut off at the mean score and dummy-coded to represent whether an adolescent had that stressor or risk (=1) or did not have it (=0). The parent and home risk scores ranged from 0 to 4 (25 youth had 0 stress/risks, 22 had 1 risk, 30 had 2 risks, 30 had 3 risks, and 12 had all 4 stressors/risks).

## 3. Results

Analyses were completed using SPSS v28 [77]. Correlational analyses were first examined to assess the direct correlations between all of the variables of interest. LEP was correlated with all of the other variables except for parent and home stressors and was only marginally correlated with child BMI percentile (*r* = 0.18, *p* = 0.052; see Table 1 for all correlations). LEP was negatively related to stressful events (MESA), depressive symptoms, aggression, and frustration, and positively related to child self-esteem. The mental health outcomes were generally correlated with one another, but none of them were correlated with child BMI percentile.

The hypotheses in the current study were partially supported in most cases. Hypothesis 1 stated that stressful life events would be related to all five of the health outcomes; Hypothesis 2 stated that parent and home stressors would be directly related to all five of the health outcomes; Hypothesis 3 stated that ethnic pride would moderate the links between stressful life events and health outcomes; and Hypothesis 4 stated that ethnic pride would moderate the links between parent and home stressors and all five of the health outcomes in Latinx youth in the Midwest.

For example, H1 was mostly supported, with MESA relating to all of the mental health outcomes but not being related to child BMI percentiles (See Table 1, Table 2, Table 3, Table 4, Table 5 and Table 6). MESA was positively related to depressive symptoms, aggression, and frustration, and negatively related to child self-esteem. In the stepwise regression models, after controlling for family income, child age, and child gender, MESA was significantly related to depressive symptoms, aggression, frustration, and self-esteem, but was not significantly related to child BMI percentiles (see Table 2, Table 3, Table 4, Table 5 and Table 6).

Hypothesis 2 was also only partially supported. PHSR was significantly and positively correlated with depressive symptoms and child aggression, and only marginally and negatively associated with self-esteem (see Table 1). After controlling for family income, child age, and child gender, PHSR significantly predicted depressive symptoms, aggression, and frustration in the stepwise regression analyses, but was not significantly related to self-esteem and was only marginally related to child BMI percentile (see Table 2, Table 3, Table 4, Table 5 and Table 6).

Hypothesis 3 focused on the potential moderating role of LEP between MESA and the health outcomes and was only partially supported. In the adjusted regression models, LEP was a significant moderator for the effects of MESA on child frustration and self-esteem, marginally related to depressive symptoms as a moderator, and not related to aggression or child BMI percentile as a moderator (see Table 2, Table 3, Table 4, Table 5 and Table 6).

Finally, Hypothesis 4 focused on the potential moderating role of LEP between PHSR and health outcomes and was not supported as a significant moderator with any of the health outcomes (see Table 2, Table 3, Table 4, Table 5 and Table 6).

## 4. Discussion

This study focused on how different types of stress, such as stressful events (MESA) and parent and home risks (PHSR), were related to internalizing and externalizing behaviors, child self-esteem, and obesity risk in a sample of rural Midwestern Latinx youth. Generally, the study found similar connections between stressors and mental health outcomes and supported previous literature linking stress to depression [2,8,9], externalizing behaviors such as aggression and frustration [10,11,12,13,14,15], and child self-esteem [9,16,17]. The study also supported prior evidence that culture is related to both stress and health outcomes [7,11,25,26,27,28,35,36].

The current study supported previous literature in relation to the associations with ethnic pride as well, showing that LEP is related to better mental health outcomes, less risky behaviors, and higher self-esteem [9,26,28,58,59,60]. These findings suggest that affiliation with, and appreciation for, one′s cultural heritage can act as an important buffer against various negative outcomes by increasing one′s sense of belonging and value. Higher ethnic pride can play an important role in building identity and confidence, which may in turn lead to protective influences in the midst of stressors and risks faced by Latinx children and adolescents.

The value of ethnic pride (LEP) for these Latinx youth was not consistently shown across the links between stressors and health outcomes, but a few of these specific links were supported and are worth mentioning. Although LEP did not moderate the influence of parent and home stressors and risks (PHSR), it was a significant moderator between stressful life events (MESA) and frustration outcomes, and MESA and self-esteem, and was marginally related to the link between MESA and depressive symptoms. These findings suggest that LEP may play an important protective role when it comes to experiencing certain life events in this population. This is particularly important as Latinx youth from rural areas and who come from families with low-income often face unique challenges and health disparities. LEP and other cultural variables that emphasize strength and value within one′s culture may be important factors to consider in intervention work to improve the health and well-being of Latinx youth and other populations.

The youth in this study navigate an interesting environment where the majority of them are growing up in homes where both parents are immigrants to the U.S. and still primarily speak Spanish, whereas the children were almost all born in the U.S., often identify as American, and primarily speak English outside the home and Spanish within the home. For many of these youth, the Spanish language and Hispanic culture are strongly promoted and emphasized by parents at home, while at school they are faced with pressures from peers to use English and fit in with the typical American culture. This can create unique pressures for acculturation and trying to maintain one′s heritage. When youth are encouraged to have confidence and pride in their cultural origin while navigating the processes of acculturation, their general well-being seems to benefit through better mental and physical health outcomes. These findings support previous literature that found links between the level of acculturation and health outcomes [9,29,32,47,49,53,63,64,65].

### 4.1. Culture, Physical Health, and the Hispanic Paradox

The lack of findings relating to child BMI percentiles was surprising to the authors, as previous studies have often noted associations between stress and health problems and obesity [2,19,22,78]. One potential reason that obesity was not related to the other variables in this sample could have been due to the high rate of obesity in this particular sample, with approximately 40.3% of the children meeting the criteria for being considered obese for their age, gender, height, and weight levels, and another 20.2% being classified as overweight. These numbers are higher than most pre-adolescent populations, and even higher than the CDC [23] report that shows the average prevalence of obesity in Latinx youth to be closer to 26.2%. Additionally, some studies actually show that obesity in Latinx families does not relate to some of the commonly seen associations with variables in other populations. This is known as the Hispanic Paradox.

A great deal of research has been done to explore the Hispanic Paradox, which is the odd relationship between socioeconomic status and health that is found in Latinx households in the United States. The paradox can be described as Hispanic/Latinx families with low socioeconomic status achieving better health outcomes than non-Latinx families with the same socioeconomic status, in terms of both physical and mental health [30,64,79,80]. Several studies have found that the validity of the paradox only applies to Latinx immigrants; in other words, the paradox does not tend to hold true for generations beyond the first and second, especially in terms of obesity and overweight [30,31,32,33,80,81]. However, some research suggests that in the case of Latinx adolescents, the Hispanic paradox applies less to first generation immigrants than to U.S. born individuals [28,48]. While it is difficult to decipher the exact cause, or causes, of the Hispanic Paradox, several studies have examined the correlations between health outcomes in Latinx individuals and elements of Latinx identity. Some researchers found that better dietary habits were associated with a greater adherence to a Latinx culture of origin [53], and others concluded that a traditional Latinx lifestyle was related to lower risk for overweight and obesity [32]. The plausibility of these relationships is supported by research that determined that, among Mexican immigrants, the Hispanic Paradox does not exist solely because the healthiest Mexicans are immigrating to the United States, but because some element of Latinx immigrants′ identity beyond their physiology is responsible for the Hispanic Paradox [82]. Multiple studies have found that the loss of Latinx culture and the adoption of U.S. culture are tied to poorer health outcomes and the annulling of the Hispanic Paradox, including increased risk for CVD [63], increased risk for obesity and overweight [30,32], and poorer dietary habits [52,53].

The existence of the Hispanic Paradox has sparked a great deal of interest among social scientists in the United States, in part because the causes of the paradox and the extent of its effects are not yet fully known. The role of Latinx culture in leading to the better-than-expected health outcomes of Latinx individuals is worth exploring, especially since theories like selective migration have been rejected by several studies [82,83] and relationships between culture and health in Latinx communities have been found. The current study was composed of Latinx youth from rural Midwestern towns, and the majority of families had very low income levels and parental education (common measures of SES).

### 4.2. Culture and Familism

Along with these cultural factors, the cultural value of familism also stands out as an important factor to consider in these moderating relationships. Familism is a cultural value among Latinx families in which individuals prioritize familial relationships and the associated mutual support to the point where family becomes an important part of one′s identity [58]. In fact, higher familism both correlated with ethnic identity [46] and predicted ethnic pride [84]. Among Latinx individuals, it would appear that this important cultural value can mediate the relationship between ethnic identity and mental health, protect against mental distress [46], and predict improved psychological functioning as well as higher self-esteem [84].

The influences of familism on mental health also appear with narrower focus on internalizing symptoms. Overall, familism appears to be negatively correlated with internalizing symptoms [9]. This relationship seems especially well documented for depression, as numerous studies report that familism is negatively associated with depressive symptoms [37,58] and negatively predicted the same [84]. In addition, reported high familism moderated the relationship between acculturative stress and depression, significantly weakening the connection [7]. It may be important for future researchers to address the role that familism plays in the development of ethnic pride and other cultural protective factors in children and adolescents.

### 4.3. Limitations, Strengths, and Future Directions

The current study is not without limitations. Although the sample size allowed the researchers to analyze the data and test the hypotheses, the sample size could have benefited from being larger. Data were also collected from self-reporting, so there is always a potential for subjective bias to influence the data if participants are not sure about a question or feel uncomfortable answering any questions honestly. The data is cross-sectional, and therefore does not allow any causal findings to be tested over time, but rather only allows for concurrent association claims to be addressed. Some language barriers may have influenced data collection as well, even though the researchers who interacted with the families were bilingual and all of the surveys and materials were available in both Spanish and English. Another limitation is the use of BMI percentile as a measure of physical health. Although this study used objective measurements of height and weight, the BMI percentile calculation itself is not as indicative of health as previous studies may have suggested. The calculations were based on differences across cultures and were sometimes used in the eugenics movement to focus on differences across race and ethnicity. BMI also does not account for weight distribution and racial differences in body type, body image preferences, central adiposity, lean muscle mass differences, and other limitations. These differences can be important to note, especially if studies are comparing weight status across racial groups.

This study also included several important strengths. One of the greatest strengths was that the entire sample included Latinx youth and parents from the Midwestern U.S. Little research attention has been given to this population, and more is needed, especially on families such as the current participants who lived in rural Midwestern locations. In addition, the researchers took a salutogenic approach to trying to address strengths within this community by focusing on cultural strengths within the study itself rather than trying to compare this sample to another ethnicity or race. All of the data were collected by participants in their own homes after building a rapport with the trained bilingual researchers. This should have helped the participants feel more comfortable with the informed consent process, collection of anthropometric measurements, and completion of surveys in their homes. Researchers also allowed participants to choose to complete all questionnaires in Spanish or English. The surveys included commonly used and validated survey items, and the child BMI percentile was measured and calculated using standard CDC recommended procedures. Participants were given instructions and allowed three days to complete the surveys. This should have helped them to not feel rushed and allowed them to complete the surveys on their own and in private. Sealed envelopes were provided to parents and children, who were instructed to complete the surveys and then seal them in the large envelopes to protect their privacy and help them to feel comfortable to report accurately on each of the measures.

Future researchers should continue to build on the links between stress and health, examining how daily hassles and stressors, life events, and stressors or risks within the home or family influence child and adolescent health over time. Longitudinal studies with more detailed cultural assessments might be helpful in identifying which aspects of culture and acculturation play important roles in increasing benefits or protecting health and well-being. More work is needed in minority populations such as this population of rural Midwestern Latinx youth and their families. These families often face increased health disparities and increased potential for stressors that may be unique, such as acculturative stress, stressors associated with language differences, lack of awareness of local resources after immigrating to a new area, lack of familial support from extended family, and other risks. For these reasons, it will be helpful to continue to identify not only the links between stressors and health outcomes in this population, but also potential strategies that could be used in targeted interventions to address positive change and improvements for other families based on the strengths and resilient factors of others in their similar ethnic, cultural, or racial groups.

## 5. Conclusions

This paper contributes to the literature by demonstrating that stressful events and stressors or risks within the home environment or relationships can impact health outcomes and self-esteem in Latinx youth. This is important to address because many Latinx youth face increased stressors and health disparities for various health outcomes, and more research is needed to address how these stressors impact health over time. The current study also addressed the importance of Latinx ethnic pride as a moderator in the link between stressors and health outcomes. The practical importance of this finding is that emphasizing or encouraging Latinx ethnic pride may be helpful for other risk factors and health as well. Maybe when adolescents feel more connected to their ancestral heritage, it provides a protective buffering factor against stressors that they face. If this is shown to be replicated in other studies, that could guide our intervention strategies in working with this population. If these findings are true in the Latinx community, then we may also examine and find ethnic pride to be an important protective factor in other cultures such as groups of refugees and immigrants. Future researchers should continue to examine the important role of ethnic pride and seek to understand the potential protective benefits that feeling closely linked to your cultural identity may play in children and adolescents as they balance becoming acculturated into the place they live while maintaining the cultural values and identity that their parents and ancestors have contributed to their identity.

## Figures and Tables

**Table 1 ijerph-19-16966-t001:** Intercorrelations of stressors, health outcomes, and culture in Latinx youth.

Variable	1	2	3	4	5	6	7	8
1—Latinx Ethnic Pride (LEP)	--							
2—Stressful events (MESA)	−0.19 *	--						
3—Parent Home Stressors (PHSR)	−0.09	0.16 ^+^	--					
4—Child Depressive Symptoms	−0.33 **	0.30 *	0.28 **	--				
5—Child Aggression	−0.27 **	0.27 **	0.26 **	0.40 **	--			
6—Child Frustration	−0.41 *	0.41 **	0.22 *	0.54 **	0.48 **	--		
7—Child Self-Esteem	0.39 **	−0.34 **	−0.20 *	−0.64 **	−0.36 **	−0.41 **	--	
8—Child BMI Percentile	0.18 ^+^	0.04	0.15	0.07	0.13	−0.01	0.00	--

Note: ^+^
*p* < 0.10, * *p* < 0.05, ** *p* < 0.01.

**Table 2 ijerph-19-16966-t002:** Moderated regression analysis of the role of ethnic pride on child depressive symptoms.

**Variable**	**Step 1**				**Step 2**				**Step 3**			
	** *b* **	** *SE* **	**β**	** *t* **	** *b* **	** *SE* **	**β**	** *t* **	** *b* **	** *SE* **	**β**	** *t* **
Family Income	−0.01	0.03	−0.05	−0.47	−0.02	0.03	−0.06	−0.63	−0.02	0.03	−0.05	−0.59
Child Age	0.01	0.11	0.01	0.10	0.05	0.10	0.05	0.51	0.03	0.10	0.03	0.32
Child Gender	0.05	0.15	−0.03	0.34	0.08	0.14	0.05	0.58	0.08	0.14	0.05	0.53
MESA					0.18	0.07	0.24	2.58	0.23	0.07	0.30	3.06
Ethnic Pride					−0.22	0.07	−0.28	−3.05	−0.24	0.07	−0.31	−3.35
MESA × Ethnic Pride									0.12	0.07	0.18	1.78
*F*	*F* (3, 103) = 0.13				*F* (5, 101) = 4.10 **				*F* (6, 100) = 4.01 **			
*F* change	Δ*F* (3, 103) = 0.13				Δ*F* (2, 10,189) = 10.02 **				Δ*F* (1, 100) = 3.12 ^+^			
*R^2^*	0.00				0.17				0.19			
*R^2^* change	0.00				0.17				0.03			
**Variable**	**Step 1**				**Step 2**				**Step 3**			
	** *b* **	** *SE* **	**β**	** *t* **	** *b* **	** *SE* **	**β**	** *t* **	** *b* **	** *SE* **	**β**	** *t* **
Family Income	−0.01	0.03	−0.05	−0.47	−0.02	0.03	−0.09	−0.95	−0.03	0.03	−0.10	−1.13
Child Age	0.01	0.11	0.01	0.10	0.02	0.10	0.02	0.23	0.02	0.10	0.02	0.23
Child Gender	0.05	0.15	0.03	0.34	0.11	0.14	0.07	0.78	0.11	0.14	0.07	0.82
Parent/Home Stressors					0.23	0.07	0.29	3.21	0.22	0.07	0.29	3.19
Ethnic Pride					−0.23	0.07	−0.30	−3.30	−0.22	0.07	−0.29	−3.17
Parent/Home Stressors × Ethnic Pride									−0.10	0.07	−0.14	−1.53
*F*	*F* (3, 103) = 0.13				*F* (5, 101) = 4.92 **				*F* (6, 100) = 4.55 **			
*F* change	Δ*F* (3, 103) = 0.13				Δ*F* (2, 101) = 12.07 **				Δ*F* (1, 100) = 2.35			
*R^2^*	0.00				0.20				0.21			
*R^2^* change	0.00				0.19				0.02			

Note: *N* = 119. ^+^
*p* < 0.10, ** *p* < 0.01.

**Table 3 ijerph-19-16966-t003:** Moderated regression analysis of the role of ethnic pride on child aggression.

**Variable**	**Step 1**				**Step 2**				**Step 3**			
	** *b* **	** *SE* **	**β**	** *t* **	** *b* **	** *SE* **	**β**	** *t* **	** *b* **	** *SE* **	**β**	** *t* **
Family Income	0.03	0.03	0.11	1.14	0.03	0.02	0.11	1.13	0.03	0.02	0.11	1.17
Child Age	−0.05	0.10	−0.05	−0.55	−0.02	0.29	−0.02	−0.20	−0.03	0.09	−0.03	−0.33
Child Gender	−0.17	0.14	−0.13	−1.28	−0.16	0.13	−0.11	−1.21	−0.16	0.13	−0.12	−1.25
MESA					0.16	0.06	0.24	2.53	0.19	0.07	0.28	2.81
Ethnic Pride					−0.14	0.07	−0.20	−2.10	−0.15	0.07	−0.22	−2.30
MESA × Ethnic Pride									0.08	0.06	0.13	1.26
*F*	*F* (3, 103) = 1.17				*F* (5, 101) = 3.48 **				*F* (6, 100) = 3.18 **			
*F* change	Δ*F* (3, 103) = 1.17				Δ*F* (2, 101) = 6.75 **				Δ*F* (1, 100) = 1.58			
*R^2^*	0.03				0.15				0.16			
*R^2^* change	0.03				0.11				0.01			
**Variable**	**Step 1**				**Step 2**				**Step 3**			
	** *b* **	** *SE* **	**β**	** *t* **	** *b* **	** *SE* **	**β**	** *t* **	** *b* **	** *SE* **	**β**	** *t* **
Family Income	0.03	0.03	0.11	1.14	0.02	0.02	0.08	0.85	0.02	0.02	0.06	0.68
Child Age	−0.05	0.10	−0.06	−0.55	−0.04	0.09	−0.04	−0.49	−0.04	0.09	−0.05	−0.49
Child Gender	−0.17	0.14	−0.13	−1.28	−0.14	0.13	−0.10	−1.04	−0.13	0.13	−0.09	−1.01
Parent/Home Stressors					0.20	0.06	0.28	3.04	0.19	0.06	0.28	3.01
Ethnic Pride					−0.15	0.06	−0.21	−2.32	−0.14	0.06	−0.20	−2.19
Parent/Home Stressors × Ethnic Pride									−0.09	0.06	−0.13	−1.42
*F*	*F* (3, 103) = 1.17				*F* (5, 101) = 4.11 **				*F* (6, 90) = 3.80			
*F* change	Δ*F* (3, 103) = 1.17				Δ*F* (2, 101) = 8.28 **				Δ*F* (1, 90) = 2.01			
*R^2^*	0.03				0.17				0.19			
*R^2^* change	0.03				0.14				0.02			

Note: *N* = 119. ** *p* < 0.01.

**Table 4 ijerph-19-16966-t004:** Moderated regression analysis of the role of ethnic pride on child frustration.

**Variable**	**Step 1**				**Step 2**				**Step 3**			
	** *b* **	** *SE* **	**β**	** *t* **	** *b* **	** *SE* **	**β**	** *t* **	** *b* **	** *SE* **	**β**	** *t* **
Family Income	−0.01	0.03	−0.05	−0.49	−0.02	0.02	−0.06	−0.74	−0.02	0.02	−0.06	−0.69
Child Age	0.02	0.11	0.02	0.21	0.09	0.09	0.08	0.90	0.06	0.09	0.05	0.65
Child Gender	−0.02	0.16	−0.02	−0.15	0.01	0.13	0.01	0.11	0.00	0.13	0.00	0.04
MESA					0.28	0.07	0.36	4.28	0.33	0.07	0.44	4.98
Ethnic Pride					−0.28	0.07	−0.36	−4.29	−0.31	0.07	−0.40	−4.76
MESA × Ethnic Pride									0.16	0.06	00.22	2.48
*F*	*F* (3, 103) = 0.09				*F* (5, 101) = 9.30 **				*F* (8, 100) = 9.16 **			
*F* change	Δ*F* (3, 103) = 0.09				Δ*F* (2, 101) = 23.04 **				Δ*F* (1, 100) = 6.14 *			
*R* ^2^	0.00				0.32				0.36			
*R*^2^ change	0.00				0.32				0.04			
**Variable**	**Step 1**				**Step 2**				**Step 3**			
	** *b* **	** *SE* **	**β**	** *t* **	** *b* **	** *SE* **	**β**	** *t* **	** *b* **	** *SE* **	**β**	** *t* **
Family Income	−0.01	0.03	−0.05	−0.49	−0.03	0.03	−0.09	−0.99	−0.03	0.03	−0.10	−1.16
Child Age	0.02	0.11	0.02	0.21	0.03	0.10	0.03	0.35	0.03	0.10	0.03	0.34
Child Gender	−0.02	0.16	−0.02	−0.15	0.04	0.14	−0.03	0.29	0.05	0.14	0.03	−0.33
Parent/Home Stressors					0.16	0.07	0.20	2.30	0.16	0.07	0.20	2.26
Ethnic Pride					−0.32	0.07	−0.41	−4.65	−0.31	0.07	−0.40	−4.53
Parent/Home Stressors × Ethnic Pride									−0.09	0.07	−0.13	−1.44
*F*	*F* (3, 103) = 0.09				*F* (5, 101) = 6.07 **				*F* (6, 100) = 5.46 **			
*F* change	Δ*F* (3, 103) = 0.09				Δ*F* (2, 101) = 15.00 **				Δ*F* (1, 100) = 2.07			
*R* ^2^	0.00				0.23				0.25			
*R*^2^ change	0.00				0.23				0.02			

Note: *N* = 119. * *p* < 0.05, ** *p* < 0.01.

**Table 5 ijerph-19-16966-t005:** Moderated regression analysis of the role of ethnic pride on child self-esteem.

**Variable**	**Step 1**				**Step 2**				**Step 3**			
	** *b* **	** *SE* **	**β**	** *t* **	** *b* **	** *SE* **	**β**	** *t* **	** *b* **	** *SE* **	**β**	** *t* **
Family Income	0.01	0.02	0.08	0.75	0.02	0.02	0.09	1.00	0.01	0.02	0.08	0.96
Child Age	0.08	0.07	0.11	1.12	0.05	0.06	0.08	0.86	0.07	0.06	0.10	1.13
Child Gender	0.08	0.10	0.08	0.79	0.05	0.08	0.05	0.60	0.06	0.08	0.06	0.69
MESA					−0.11	0.04	−0.24	−2.76	−0.15	0.04	−0.32	−3.49
Ethnic Pride					0.18	0.04	0.39	4.38	0.20	0.04	0.42	4.84
MESA × Ethnic Pride									−0.10	0.04	−0.23	−2.42
*F*	*F* (3, 101) = 0.84				*F* (5, 99) = 7.25 **				*F* (6, 98) = 7.31 **			
*F* change	Δ*F* (3, 101) = 0.84				Δ*F* (2, 99) = 16.48 **				Δ*F* (1, 98) = 5.85 *			
*R^2^*	0.02				0.27				0.312			
*R^2^* change	0.02				0.24				0.04			
**Variable**	**Step 1**				**Step 2**				**Step 3**			
	** *b* **	** *SE* **	**β**	** *t* **	** *b* **	** *SE* **	**β**	** *t* **	** *b* **	** *SE* **	**β**	** *t* **
Family Income	0.01	0.02	0.08	0.75	0.02	0.02	0.11	1.20	0.02	0.02	0.11	1.22
Child Age	0.08	0.07	0.11	1.12	0.07	0.06	0.10	1.13	0.07	0.06	0.10	1.12
Child Gender	0.08	0.10	0.08	0.79	0.04	0.09	0.04	0.49	0.04	0.09	0.04	0.49
Parent/Home Stressors					−0.06	0.04	−0.13	−1.40	−0.06	0.04	−0.13	−1.38
Ethnic Pride					0.20	0.04	0.42	4.68	0.20	0.04	0.42	4.62
Parent/Home Stressors × Ethnic Pride									0.01	0.04	0.02	0.25
*F*	*F* (3, 101) = 0.84				*F* (5, 99) = 5.82 **				*F* (6, 98) = 4.82 **			
*F* change	Δ*F* (5, 101) = 0.84				Δ*F* (2, 99) = 12.99 **				Δ*F* (1, 98) = 0.06			
*R^2^*	0.02				0.23				0.23			
*R^2^* change	0.02				0.20				0.00			

Note: *N* = 119. * *p* < 0.05, ** *p* < 0.01.

**Table 6 ijerph-19-16966-t006:** Moderated regression analysis of the role of ethnic pride on child BMI percentile.

**Variable**	**Step 1**				**Step 2**				**Step 3**			
	** *b* **	** *SE* **	**β**	** *t* **	** *b* **	** *SE* **	**β**	** *t* **	** *b* **	** *SE* **	**β**	** *t* **
Family Income	0.28	0.76	0.03	0.37	0.39	0.75	0.05	0.52	0.37	0.75	0.05	0.50
Child Age	1.60	2.94	0.05	0.54	1.98	2.92	0.06	00.68	2.29	2.94	0.07	0.78
Child Gender	−15.76	4.15	−0.35	−3.80	−16.51	4.09	−0.37	−4.04	−16.39	4.10	−0.37	−4.00
MESA					2.17	2.01	0.10	1.08	1.45	2.15	0.07	0.68
Ethnic Pride					4.56	2.05	0.21	2.22	4.91	2.09	0.22	2.35
MESA × Ethnic Pride									−1.95	2.02	−0.10	−0.97
*F*	*F* (3, 103) = 5.15				*F* (5, 101) = 4.26 **				*F* (6, 100) = 3.70 **			
*F* change	Δ*F* (3, 103) = 5.15 **				Δ*F* (2, 101) = 2.67 ^+^				Δ*F* (1, 100) = 0.93			
*R^2^*												
*R^2^* change												
**Variable**	**Step 1**				**Step 2**				**Step 3**			
	** *b* **	** *SE* **	**β**	** *t* **	** *b* **	** *SE* **	**β**	** *t* **	** *b* **	** *SE* **	**β**	** *t* **
Family Income	0.28	0.76	0.03	0.37	0.27	0.74	0.03	0.36	0.16	0.74	0.02	0.22
Child Age	1.60	2.94	0.05	0.54	1.67	2.86	0.05	0.59	1.66	2.86	0.05	0.58
Child Gender	−15.76	4.15	−0.35	−3.80	−16.11	4.05	−0.36	−3.98	−16.00	4.04	−0.36	−3.96
Parent/Home Stressors					3.82	2.04	0.17	1.88	3.75	2.03	0.17	1.84
Ethnic Pride					4.56	2.00	0.21	2.28	4.78	2.01	0.22	2.38
Parent/Home Stressors × Ethnic Pride									−2.29	1.90	−0.11	−1.20
*F*	*F* (3, 103) = 5.15				*F* (3, 101) = 4.82 **				*F*(3, 100) = 4.28 **			
*F* change	Δ*F* (3, 103) = 5.15 **				Δ*F* (2, 101) = 3.89 *				Δ*F*(1, 100) = 1.45			
*R^2^*	0.13				0.19				0.20			
*R^2^* change	0.13				0.06				0.01			

Note: *N* = 119. ^+^
*p* < 0.10, * *p* < 0.05, ** *p* < 0.01.

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
