# Peer review of "Stress and Health Outcomes in Midwestern Latinx Youth: The Moderating Role of Ethnic Pride"

_ijerph, 2022, doi:10.3390/ijerph192416966_

Round 1

Reviewer 1 Report

Overall this is an interesting article and research worth publishing with edits.

General edits:

P2 L 28 adolescence

P2 L 42-45In times of chronic  stress, keeping these systems activated for longer periods of time has proven to be harmful  on physical health because of chronically elevated blood pressure and a weakened or damaged immunity due to increased exposure to stress hormones [2].- worded oddly.

P3 L45 Define Acculturation and Acculturative Stress

P13 L4 negatively

Remind readers in your results what your Hypotheses were.

P16 L17 this population

Regarding limitations, while the BMI is still used and cited by the CDC it is also noted that the BMI is a tool born out of the eugenics movement and ergo, inherently racist.  Lambert Adolphe Jacques Quetelet was a statistician and formulated to BMI as part of a project to determine “the average man” and compared non-average measurements (found in black individuals) as contrary to the “normal” of whites. Your study specifically focuses on Hispanic individuals and focuses on the use of the BMI (which also does not accurately measure adiposity) and therefore must be disclosed.

Thank you 

Author Response

Overall this is an interesting article and research worth publishing with edits.

     ~Response: Thank you for your careful review and consideration.

General edits:

P2 L 28 adolescence

     ~Response: We fixed this. Thank you for catching this mistake.

P2 L 42-45In times of chronic stress, keeping these systems activated for longer periods of time has proven to be harmful on physical health because of chronically elevated blood pressure and a weakened or damaged immunity due to increased exposure to stress hormones [2].- worded oddly.

     ~Response: Thank you. We rewrote this sentence to make it easier to read.

P3 L45 Define Acculturation and Acculturative Stress

     ~Response: Thank you. We added these general definitions to page 3 (L45-46) and 4 L1-2).  

P13 L4 negatively

     ~Response: We fixed this typo.

Remind readers in your results what your Hypotheses were.

     ~Response: Thank you. We added the hypotheses to the results to remind the reader what each of the 4 specific hypotheses were in this study.

P16 L17 this population

      ~Response: Thank you. We corrected this as well.

Regarding limitations, while the BMI is still used and cited by the CDC it is also noted that the BMI is a tool born out of the eugenics movement and ergo, inherently racist.  Lambert Adolphe Jacques Quetelet was a statistician and formulated to BMI as part of a project to determine “the average man” and compared non-average measurements (found in black individuals) as contrary to the “normal” of whites. Your study specifically focuses on Hispanic individuals and focuses on the use of the BMI (which also does not accurately measure adiposity) and therefore must be disclosed.

     ~Response: Thank you for noting this. We agree that BMI percentile is not always the most descriptive measure of weight status and has been used inappropriately at times. It is a very common measure when discussing the health challenges associated with weight status. In this population and study, BMI was not a significant finding but we still included it as a proxy variable for physical health. We added the limitation to the discussion as follows, Another limitation is the use of BMI percentile as a measure of physical health. Although this study used objective measurements of height and weight, the BMI percentile calculation itself is not as indicative of health as previous studies may have suggested. The calculations were based on differences across cultures and were sometimes used in the eugenics movement to focus on differences across race and ethnicity. BMI also does not account for weight distribution and racial differences in body type, body image preferences, central adiposity, lean muscle mass differences and more limitations. These differences can be important to note, especially if studies are comparing weight status across racial groups.”

Reviewer 2 Report

Thank you for giving me the possibility to review this paper. It is an intriguing topic of interest since it focuses on the interrelation between environmental stressors, mental health, and cultural heritage.

The authors supported hypotheses with the literature background, and the methodological design is well prepared and described. My only remark is about the sample size, which seems too small for such a study. I suggest the authors use the Gpower test to report the adequate sample size for their research.

Nevertheless, the results are well explained, even if the discussion could better describe the current study's social implications. Why is this study critical from a psychosocial framework? I guess the Discussion could write the practical impact of the study. 

Author Response

Thank you for giving me the possibility to review this paper. It is an intriguing topic of interest since it focuses on the interrelation between environmental stressors, mental health, and cultural heritage.

     ~Response: Thank you for your careful review and consideration.

The authors supported hypotheses with the literature background, and the methodological design is well prepared and described. My only remark is about the sample size, which seems too small for such a study. I suggest the authors use the Gpower test to report the adequate sample size for their research.

     ~Response: Thank you for this excellent question. The researchers did in fact conduct a power analysis a priori to data collection. Using G*Power 3.1.7, the researchers did a calculation to attain the needed sample size when they were designing the study. Based on a medium effect size of f2= 0.15, an alpha = .05, and a standard power needed of 0.80, the sample size was estimated to need at least 98 participants. We also confirmed this number on the danielsoper.com website where you can calculate needed sample size for this type of analysis. On his website it estimated that we needed a sample size of 97 (given the same effect size of .15, alpha of .05, and power of .80)

       Our goal was to collect a larger number and we were able to get up to 119 families, which was more than the needed sample size for our estimates. This was a very difficult population to get a larger sample size from as it was primarily rural Latinx families living in the Midwest. We used connections with local school superintendents, county Extension educators, and every other resource we could think of (e.g., flyers, attending local sporting and music events, snowball/referral sampling, etc.) to reach more potential participants. Our research staff were all bilingual, including some native Spanish speakers, to try to connect with the families.

Nevertheless, the results are well explained, even if the discussion could better describe the current study's social implications. Why is this study critical from a psychosocial framework? I guess the Discussion could write the practical impact of the study. 

     ~Response:  We added some practical implications of the findings to the discussion in the conclusion paragraph. Thank you for this suggestion.